# Circulating microRNAs as a Prognostic Tool to Determine Treatment Efficacy in Lung Cancer Patients Undergoing Pembrolizumab PD-1 Blockade Immunotherapy

**DOI:** 10.3390/cancers16244202

**Published:** 2024-12-17

**Authors:** Mishfak A. Mohamed Mansoor, Xiang Zhu, Sarah Aslam Ashiqueali, Md Tanjim Alam, Hanna Winiarska, Pawel Pazdrowski, Filip Kaminski, Alicja Copik, Michal M. Masternak, Barbara Kuznar-Kaminska

**Affiliations:** 1Burnett School of Biomedical Sciences, College of Medicine, University of Central Florida, Orlando, FL 32827, USA; mishfak@ucf.edu (M.A.M.M.); xiang.zhu@ucf.edu (X.Z.); sarah.ashiqueali@ucf.edu (S.A.A.); mdtanjim.alam@ucf.edu (M.T.A.); alicja.copik@ucf.edu (A.C.); michal.masternak@ucf.edu (M.M.M.); 2Department of Pulmonology, Allergology and Respiratory Oncology, Poznan University of Medical Sciences, 61-701 Poznan, Poland; winiarskahanna@gmail.com; 3Medical Faculty, Poznan University of Medical Sciences, 61-701 Poznan, Poland; pawel.pazdrowski@gmail.com (P.P.); filipkaminski04@gmail.com (F.K.); 4Department of Head and Neck Surgery, Poznan University of Medical Sciences, 61-701 Poznan, Poland

**Keywords:** lung cancer, microRNAs, cancer prognosis, PD-1 blockade, cancer immunotherapy

## Abstract

miRNAs have been shown to have altered expressions during cancer disease states, and monitoring their levels in patients undergoing cancer therapies holds potential in finding possible prognostic markers and therapeutic targets. We determined the impact of a PD-1 blockade therapy on miRNA levels, which is correlated with patients’ sex and PD-L1 status. Our findings elucidate that there was a significant decrease in a milieu of miRNA levels in the patient sera post treatment with pembrolizumab.

## 1. Introduction

With approximately 2.20 million new cases and 1.79 million fatalities annually, lung cancer (LC) stands as one of the most prevalent cancers globally, and has the highest mortality [1,2]. Despite the substantial strides made in understanding the areas of cancer biology, deploying predictive biomarkers, and refining treatment modalities, the current practice of diagnosis, treatment, and prognosis remains invasive, expensive, and often results in delayed responses [3,4].

Depending on the stage of the disease, some of the available treatments for LC include surgical removal of the tumor, radiation, and systemic therapy. Considering that most LC cases are diagnosed at an advanced stage, developing successful systemic treatment is challenging. Among the available systemic strategies, immunotherapy has emerged as the front-runner since it has shown sustained remission in the patients who respond to it [5].

Although different types of immunotherapies for LC have proven to be challenging, PD-1 (programmed cell death 1) blockade, an immune checkpoint blockade strategy, has demonstrated promising therapeutic responses [6]. Briefly, PD-1 is a surface protein found on T cells, and when bound to PD-L1 (PD-1 ligand), it keeps T cells in check, preventing them from killing other cells. Therefore, blocking PD-1 with an anti-PD-1 allows the T cells to kill more cancer cells. In 2016, the FDA approved Pembrolizumab in 2016 for patients with PD-L1 expression of more than 50%, after clinical trials concluded that the treatment had improved efficacy and side-effects were acceptable [7,8,9]. Regardless of this significant advancement, a gap still remains in determining molecular patterns between successful and failed responses to pembrolizumab treatment. Therefore, current research efforts seek to further distinguish other molecular factors that determine a good response to treatment and those that constitute a risk factor for therapeutic failure in PD-L1-positive patients.

In this clinical study, we sought to determine the changes in microRNA (miRNA) expression levels in the blood sera of patients diagnosed with LC and subjected to pembrolizumab. We aimed to evaluate their potential value as prognostic tools and the possibility of using them to monitor pembrolizumab treatment efficacy.

miRNAs are small noncoding RNAs of about 20–25 nucleotides that can target messenger RNAs (mRNAs) and suppress the gene expression by degrading mRNA copies or inhibiting translation to functional protein [10,11]. miRNAs play a vital role in the development of cancer as they regulate the expression of various tumor suppressor genes and oncogenes, emerging as important biomarkers for cancer diagnosis and prognosis [10,12,13]. While it is well established that miRNAs are critical in cancer development and various studies have explored their diagnostic value by evaluating their levels in cancer patients compared to healthy individuals, changes in miRNA levels in patients undergoing cancer treatments remain largely unexplored.

Our clinical study found that a panel of miRNAs that were previously implicated in cancers had altered expression levels post treatment with four cycles of pembrolizumab. Our survival analysis revealed that miR217 and let-7a can be used as predictive markers of patient survival. Our findings provide important insights into developing a comprehensive miRNA panel for measuring LC prognosis, pembrolizumab treatment efficacy, and predicting patient survival.

## 2. Materials and Methods

### 2.1. Study Cohort

The research was approved by the bioethics committee at the Poznan University of Medical Sciences 385/23. Informed consent was obtained from all patients.

A total of 38 patients with non-small-cell lung cancer (NSCLC) who were treated with PD-L1 inhibitors as either a monotherapy or combined with chemotherapy were recruited for this study in Poland in 2023. The inclusion criteria were as follows: a/patients > 18 years old, who signed the informed consent form, male and female; b/histopathologically or cytologically confirmed diagnosis of NSCLC; c/stage IV or III (not suitable for radical treatment, i.e., surgery or chemo-radiotherapy) cancer according to UICC-TNM 8th edition; d/negative status of epidermal growth factor receptor (EGFR), anaplastic lymphoma kinase (ALK), c-Ros oncogene -1 (ROS); e/no symptomatic metastases in the central nervous system (CNS) or futures of progression in CNS after local treatment (surgery, radiotherapy); f/the Eastern Cooperative Oncology Group (ECOG) performance status score 0–1; g/measurable lesion according to the evaluation criteria for the efficacy of solid tumors (RECIST Version 1.1).

We collected the following data: demographics, smoking history, histology, stage of the disease, overall survival (OS), and programmed death-ligand 1 (PD-L1) status, which were reported as part of the tumor proportion score (TPS) and classified as either negative (<1%), low-positive (1–49%), or positive (≥50%).

The demographics of the cohort are presented in Table 1. Further information on the cohort’s mutational status, PD-L1 status, concurrent treatments, and comorbidities is listed in the Appendix A. Briefly, the patients had several comorbidities including hypertension (65%), heart abnormalities (21%), pulmonary embolism (8%), heart failure (18%), diabetes (13%), renal failure (18%), COPD (23%), atherosclerosis (10%) and other disease states. Some patients were undergoing concurrent treatments as well, such as proton pump inhibitors (63%), statins (37%), ACE inhibitors (34%), diuretics (37%), anticoagulants (71%), NSAIDs (21%), megestrol (26%), and other treatments.

### 2.2. Experimental Methods

Blood was drawn from patients, and sera were collected before the pembrolizumab treatment started. The patients went through four cycles of treatment in 90 ± 14 days, and then a second serum sample was collected. A total of 17 patients were not able to complete the 90-day duration due to either clinical worsening or death, in which cases the second serum sample was not collected. The sera were stored in a −80 °C freezer and transported on dry ice when necessary.

Total RNA from the serum was isolated using Trizol and a commercial column purification system (miRNeasy Mini Kit, Qiagen, Valencia, CA, USA) following the manufacturer’s instructions. RNA concentration and purity were measured spectrophotometrically (absorbance at 260, 230, and 280 nm) using the Epoch™ Microplate Spectrophotometer (BioTek, Winooski, VT, USA).

To measure miRNA expression, miRNA was synthesized using the TaqMan^®^ Advanced miRNA cDNA Synthesis Kit (Applied Biosystems, Foster City, CA, USA), according to the manufacturer’s instructions. For the RT-qPCR reaction, a 1:10 dilution of cDNA template was prepared. RT-qPCR was performed using the Quantstudio™ 7 Flex System. We used TaqMan^®^ Advanced miRNA Assays (Applied Biosystems, Foster City, CA, USA) with primers specific for selected miRNA: miR-146a-5p (478399_mir), miR-126-5p (477888_mir), miR-34c-5p (478052_mir), miR-217-5p (478773_mir), miR-378c-5p (478864_mir), miR-96-5p (478215_mir), miR-149-5p (477917_mir), miR-133a-3p, let-7a miR3615, miR4516, miR16, miR181b-5p, miR20b-5p, and miR106b-5p. We selected these types of miRNA based their involvement in cancer development in previous studies conducted in our lab, as well as in the published literature [10].

The −∆∆Ct method was used to calculate and normalize the expression of each miRNA, in which −∆∆Ct was calculated as follows:
 −∆∆Ct = average (∆Ct before treatment) − ∆Ct after treatment
where ∆Ct = the Ct target gene − the Ct housekeeping gene.

The housekeeping gene was U6. In the above calculation, −∆∆Ct > 0 denotes upregulated expression, whereas −∆∆Ct < 0 denotes downregulated expression relative to the miRNA level before treatment.

### 2.3. Statistical Analysis

Among a total of 1534 expression data points acquired via qRT-PCR analysis, 154 were undetermined (Ct > 40, non-detection rate). When plotted against the mean ∆Ct, we see that the non-detection rate increases with the increase in mean ∆Ct (Figure 1A). This suggests that the non-detection occurrence is not random, and should be imputed. Conventional imputation of these non-detection rates by assigning a value of 40 Ct resulted in a bias where the data skewed to low expression (Figure 1B). We implemented the expectation–maximization algorithm to reduce this bias, which resulted in the better imputation of the non-detection rates (Figure 1C).

We performed paired *t*-tests for each sample expression change post treatment; we only included patients with data points both before and after treatment. Then, we performed two-way repeated measure ANOVAs separated by sex to evaluate changes with treatment.

Spearman’s correlation and coefficient was determined to evaluate the correlation among the miRNAs.

For survival analysis, we initially performed a log-rank test to evaluate the difference between male and female patients, with PD-L1 statuses < 50% and >50%. Then, using a multiple-variable Cox regression model, we performed a stepwise regression analysis to identify the miRNAs associated with patient survival.

## 3. Results

### 3.1. Pembrolizumab Treatment Alters the Expression of miRNAs in Both Male and Female Lung Cancer Patients

The RNA extracted from the sera of 38 patients before the immunotherapy treatment were subjected to RT-PCR on day 90, after four cycles of treatment with pembrolizumab, to determine the levels of circulating miRNAs. Our analysis indicated that the treatment caused a significant decrease in blood-circulating miRNA levels. Post treatment with pembrolizumab, we see a 4-fold decrease in miR126-5p levels; a 5-fold decrease in let-7a levels; a 4-fold decrease in miR133a-3p levels; a 2-fold decrease in miR3615 levels; a 3-fold decrease in miR4516 levels; a 3-fold decrease in miR16 levels; a 2-fold decrease in miR34c-5p levels; a 3-fold decrease in miR181b-5p levels; a 5-fold decrease in miR20b-5p levels; a 5-fold decrease in miR106b-5p levels; and a 5-fold decrease miR146-5p levels in the sera of the patients (Figure 2A,B). However, the expression level one of the measured miRNAs, miR378c, showed a nonsignificant trend toward upregulation in response to treatment (*p* = 0.138) (Figure 2A). The analysis also indicated a sex-dependent difference in the expression levels of miR133a-3p and miR4516. We analyzed the miRNA changes separately for females and males, which revealed that the miRNA changes in females are more significantly altered than in males (Table 2, Appendix A). A univariate survival analysis using the Kaplan–Meier survival curve showed a higher survival rate for females when compared with male LC patients (Figure 2C, Table 3). The data points were spread through this range, and most samples followed the same trend towards downregulation except for miR378c (Figure 3 and Figure 4A). There was no clustering of any prominence between patients who followed through the 90-day treatment (blue and green dots) and patients who could not complete the full 90 days (red dots) (Figure 3 and Figure 4A). Additionally, Spearman’s correlation coefficients between all miRNA pairs were calculated, and we found that all miRNAs were positively correlated with each other (Figure 5).

### 3.2. Impact of PD-L1 Expression Level and miRNAs on Patient Survival

PD-L1 status, widely used as a qualification marker for PD-1 blockade therapy, indicated that patients with more than 50% PD-L1 expression in tumor tissue had higher survival chances than those with PD-L1 expression that was less than 50% (Figure 4B, Table 3).

Finally, stepwise Cox regression analysis was performed to identify the miRNAs that can be used to predict patient survival. Among the 13 tested miRNAs, we identified 2 miRNAs that can predict patient survival. Let-7a was positively associated with patient survival, with high levels of expression indicating reduced risk. At the same time, miR217 was found to be negatively associated with the survival of the patients, with high levels of expression predicting an increased risk of death (Table 3). This is further apparent in a univariant Kaplan–Meier survival plot that showed a similar trend for miR217 and let-7a (Appendix A).

## 4. Discussion

Although recent studies have advanced our understanding of the roles of a wide range of microRNAs in cancer development and their potential as diagnostic markers, limited information remains on how they respond to different cancer treatments. It is critical to understand such changes as they could reveal potential miRNA-based prognostic tools for patients undergoing treatment. Additionally, the therapeutic potential of these miRNAs, determined by the pattern of changes to treatment, can be further explored and easily mimicked by developing miRNA-loaded cargos.

Our study investigates the changes in the expression of circulating miRNAs in response to pembrolizumab treatment, which is a type of PD-1 blockade immunotherapy. By unraveling miRNA changes in response to treatment, we propose a molecular-based method to monitor the response to pembrolizumab treatment which could further be used to qualify patients for immunotherapy [8,14]

Our study showed that among the panel of 13 miRNAs we tested, 12 were downregulated after treatment with pembrolizumab, while 1 miRNA showed an opposing trend. Long-term investigations are needed to determine whether the downregulation of some miRNAs leads to a point beyond which immunotherapy provides no further benefits or if continued immunotherapy is needed once certain levels of these miRNAs—associated with poor prognosis—are reached.

One of the miRNAs that was significantly suppressed after immunotherapy is miR-126. Importantly, previous studies reported that miR126-5p can regulate the proliferation and growth of LC cells by regulating STAT3 activation, [15] while overexpression of miR126-5p in LC cells enhanced growth, migration, and invasion by targeting multiple signaling pathways including the PI3K/AKT/mTOR pathway, VCAM-1, and CCR-1 [15,16,17,18]. This suggests that the downregulation of miR-126 in the serum of LC patients after pembrolizumab treatment could predict beneficial changes toward delayed migration and limited growth of LC cells in response to immunotherapy.

Recently, Yang et al. reported that another type of miRNA from our panel, miR133a-3p, is upregulated in LC and that overexpressed serum exosomal miR133a-3p promoted proliferation, cell migration, and invasion via targeting and downregulating SIRT1 [19]. Our findings further confirm that there were high levels of miR133a-3p in LC patients before pembrolizumab treatment, and that these levels were significantly reduced post treatment, which can be a positive outcome of the treatment as downregulation of this miRNA would reduce tumor growth and metastasis.

Importantly, in our initial experimental design, we planned on using miR16 as the housekeeping miRNA for our analysis; however, we noticed that miR16 was significantly affected by pembrolizumab immunotherapy. Also, previous reports indicate that miR16 is implicated in LC development. Chen et al. reported that miR16 is downregulated in LC cell lines and regulates MEK-1 levels [20]; yet, interestingly, miR-16 is further suppressed in response to immunotherapy. This further shows that miRNAs such as miR16 have larger implications on regulating tyrosine kinases and ERK/MAPK pathway proteins such as JNK and MEK1 [20].

Similarly, previous studies have shown that miR34c has been only marginally expressed in LC cells, and overexpression results in tumor suppression [21,22]. However, our findings show that not only are these miRNAs detectable in the serum of LC patients, but they also further decrease post pembrolizumab treatment. This could decrease reactive oxygen species levels in the LC cells and reduce ER stress, thereby promoting tumor development.

Importantly, miR181b-5p levels are upregulated in LC cell lines, leading to the downregulation of E-cadherin, subsequently increasing cancer cell invasion and leading to metastasis of LC [23]. We found that miR181b-5p levels were significantly reduced in LC patients who received four cycles of pembrolizumab, which could potentially restore E-cadherin levels and reduce cancer cell migration and metastasis, showing a positive outcome for pembrolizumab treatment.

Another miRNA, miR20b-5p, which was significantly downregulated by pembrolizumab treatment, was shown in other studies to be highly expressed in LC and targeting B cell translocation gene 3 (BTG3), thus downregulating BTG3 expression and leading to increased proliferation and cancer cell migration [24]. This could support the findings that reduced levels of miR20b-5p in response to immunotherapy could rescue and restore BTG3 expression and reverse cancer cell migration and metastasis, showing a positive outcome for PD-1 blockade immunotherapy.

Very similar to miR20b-5p, miR106b-5p has also been shown to be upregulated in LC and to directly play a role in the downregulation of BTG3, leading to the inhibition of cancer cell apoptosis and promoting cell proliferation [25]. Our findings show that miR106b-5p levels were reduced post treatment with pembrolizumab, and these reduced levels could promote BTG3 expression that could lead to the apoptosis of cancer cells and reduce tumor growth.

Most of the reports on the role of miR146a-5p indicate its tumor-suppressive potential and have shown it to be largely downregulated in LC cells [26,27,28]. Overexpression of miR146a-5p is associated with reduced proliferation and increased apoptosis [26,28]. However, results from our study show that LC patient sera have detectable levels of miR146a-5p, which are significantly reduced post treatment. Initially, this could suggest that significant downregulation of miR-146a could lead to increased proliferation through the inhibition of apoptosis. However, our published study indicated that miR-146a represents a significant marker of cellular senescence; moreover, our in vitro and in vivo experiments indicated that higher levels of miR-146a can cause systemic inflammation and accelerated cellular senescence, a process which promotes tumorigenesis [29]. Additionally, Tan et al. have shown that high levels of miR146a-5p are associated with poor prognosis for cancer patients, implying that our findings could mean a beneficial outcome for the PD-1 blockade treatment [30].

While the role of miR3615 has not been extensively studied in lung cancer, it has been reported as one of the miRNAs that can be used as a diagnostic signature to identify early-stage LC [31]. It has also been shown to be heavily upregulated in hepatocellular carcinoma and promote cancer progression [32]. We have shown here that miR3615 levels decrease post treatment with pembrolizumab in patients with LC, which could potentially delay cancer metastasis.

We investigated miR4516 because it is upregulated in patients with pulmonary fibrosis, which can lead to cancer development [33]. Downregulated levels of miR4516 after treatment with pembrolizumab in our study implies reduced lung damage in treated patients.

With these miRNA changes, we believe there are potential markers for patient survival; our Cox regression analysis with stepwise regression revealed that, in addition to being downregulated post treatment with pembrolizumab, both miR217 and let-7a are associated with patient survival. While miR217 was negatively associated with patient survival, indicating that higher levels would pose more risk, let-7a was positively associated with patient survival, where higher levels would cause less risk. Our findings show a favorable outcome for the treatment regarding miR217, where it is downregulated post treatment. However, our results suggest a less favorable outcome for treatment regarding let-7a, which becomes downregulated after treatment, indicating a higher risk for mortality.

Importantly, our findings about miR217 contradict previous reports of its role in LC. It has been shown that miR217 inhibits cancer progression by targeting SIRT1 and regulating the SIRT1-mediated P53/KAI1 pathway [34]. Similarly, let-7a has also been shown to be downregulated in LC, and its overexpression inhibits cancer cell migration and proliferation by cyclin D1 regulation [35]. However, our stepwise regression supports this, as higher levels of let-7a are associated with better survival. Despite these opposing outcomes, these findings add value as a potential biomarker for treatment qualification, as they could predict patient survival with noninvasive blood sera tests, moving away from the invasive biopsy required for PD-L1 status determination.

Additionally, our analysis focused on blood-circulating miRNAs, while some of the LC functional validations in the literature focused on the expression of these miRNA types in lung-cancer-specific cell lines or in cancer tissues.

One type of miRNA that showed an opposing trend to all the other miRNA types was miR378c; before treatment, there were low levels of miR378c in the serum, and these were upregulated after treatment with pembrolizumab, albeit not significantly. Our previous studies have shown that miR378c is significantly downregulated in patients with head and neck cancer [10,36]. The role of miR378c in the lungs is not widely studied, and the functional role it plays in cancer development is unclear. Further investigations into its role in lung cancer could reveal its function and its therapeutic potential.

While our miRNA panel was selected based on their role in the development of lung cancer, some were also implicated in PD-1/PD-L1 pathway regulation and T cell activation. Both miR16 and miR106b are reported to target and inhibit PD-L1 mRNA directly [37,38]. MiR106b also inhibits PTEN expression along with miR20b [39]. miR181b inhibits c-FOS, which is an activator of PD-L1 promoter, and therefore suppresses PD-L1 expression [40]. miR146a inhibit STAT1 which is upstream of PD-L1 promoter activation [41]. Cumulatively, these miRNAs could potentially lead to low PD-L1 expression, and treatment with pembrolizumab decreases these miRNA levels, leading to increased PD-L1 expression, thus enhancing patients’ responses to immunotherapy. It could also be speculated that with the changes in the expression of detected miRNAs, PD-L1 expression increases as a feedback mechanism to outcompete the pembrolizumab antibodies. However, the interplay between PD-L1 expression and miRNA in response to the PD-1 blockade needs further investigation to provide insight into fine-tuning miRNA levels to elicit a better response to immunotherapy. It is to be noted that we need to investigate this axis, as the miRNAs that are downregulated play a role in T cell activation, and PD-1 blockade-assisted T cell activation could lead to downregulation of miRNAs that are implicated in playing inhibitory roles in T cell activation.

Our study had some unavoidable limitations, particularly the cohort size. Since a significant number of patients could not complete the four cycles of pembrolizumab treatment and, therefore, were not available for post-90-day sera collection, we had to reduce our sample size for most of our data analysis. The impact of this is also shown in our survival analysis, since certain patients died or had to stop the treatment due to clinical worsening. We also acknowledge that to avoid the risk of nonresponse to immunotherapy for patients with a PD-L1 status less than 50%, we administered chemotherapy in combination with pembrolizumab. The demographic of the selected cohort is not very diverse in terms of race and geographical location. A larger cohort with more diverse subjects can produce more meaningful data. We also acknowledge that there are more sensitive miRNA-based methods that are being developed [42]. However, they remain unavailable to the general population, and as these methods develop to become more commonplace, it is crucial to keep identifying biomarker changes using available methods to better incorporate them into a sophisticated detection tool.

## 5. Conclusions

Our study provides a comprehensive miRNA panel that can be used to monitor disease prognosis for LC patients undergoing PD-1 blockade immunotherapy and to predict patient survival. We believe our findings could direct the current practice in lung cancer treatment and immunotherapy toward a noninvasive blood-test-based diagnosis, treatment qualification, and prognosis, as well as miRNA-based therapeutics for lung cancer.

## Figures and Tables

**Figure 1 cancers-16-04202-f001:**
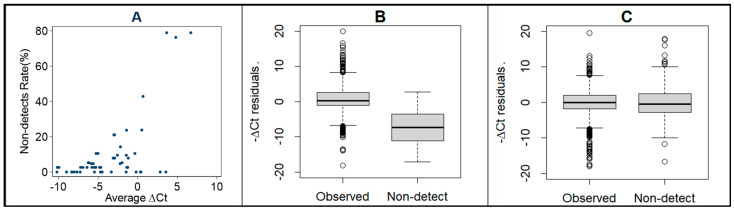
Scatter plot (**A**) indicates that the non-detection (undetermined) rate increases with the average ∆Ct by plate number, treatment, and gene. The boxplots show the comparison of −∆Ct residues between observed expressions and non-detection rates (**B**) if the conventional imputation of setting the Ct of each non-detection rate to 40 was conducted, and (**C**) if the expectation–maximization (EM) algorithm was used to impute the Ct value of each non-detection rate.

**Figure 2 cancers-16-04202-f002:**
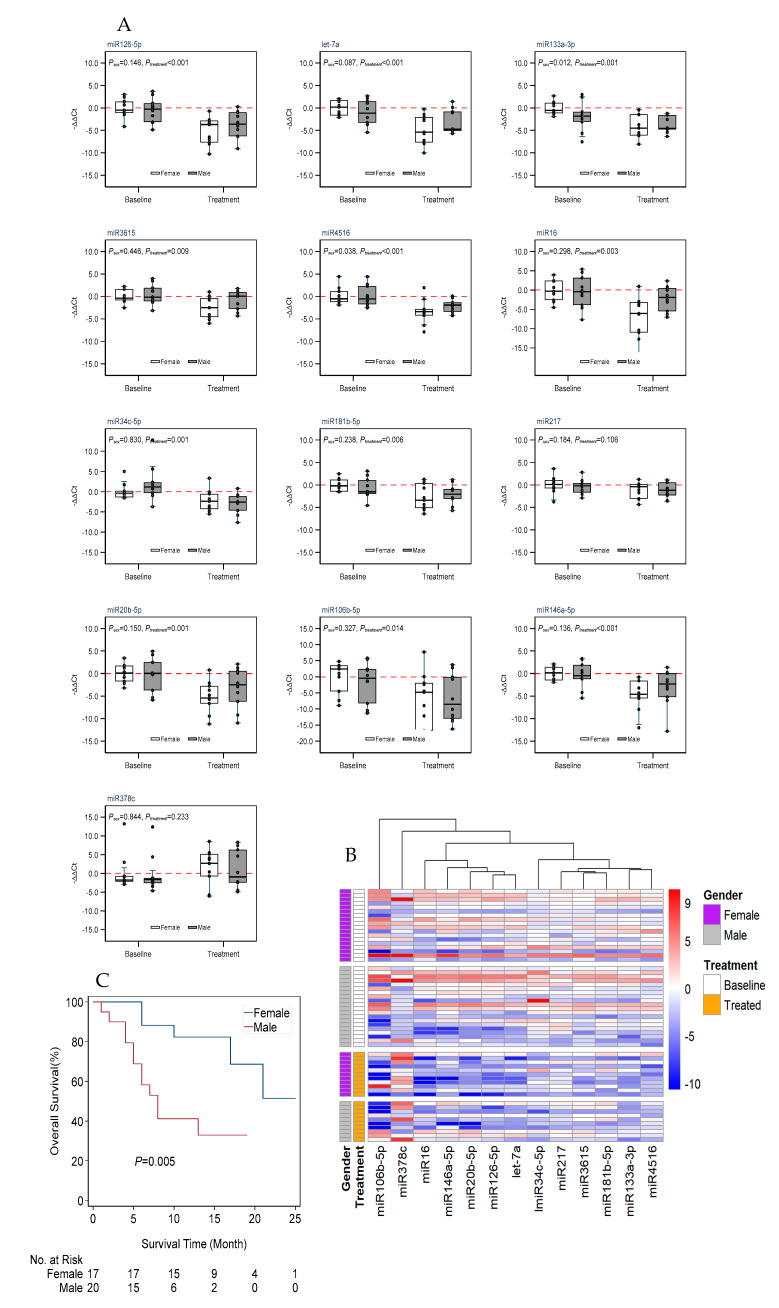
miRNA changes in the sera of LC patients treated with pembrolizumab. (**A**). Boxplots showing miRNA expression changes by treatment and gender. The 2−ΔΔCt method was used to calculate and normalize the gene expression, in which −∆∆Ct was calculated as follows: −∆∆Ct = average (∆Ct control) − ∆Ct sample, and ∆Ct = Ct _target gene_ − Ct _housekeeping gene_. The housekeeping gene was U6. Female baseline samples were set as control samples for the normalization of gene expression. The baseline expression level is indicated in red dotted line. (**B**). Heatmap showing miRNA expression profile by treatment and sex. (**C**). Kaplan–Meier plot showing the overall survival rate of patients from diagnosis to the last contact; comparison between females and males.

**Figure 3 cancers-16-04202-f003:**
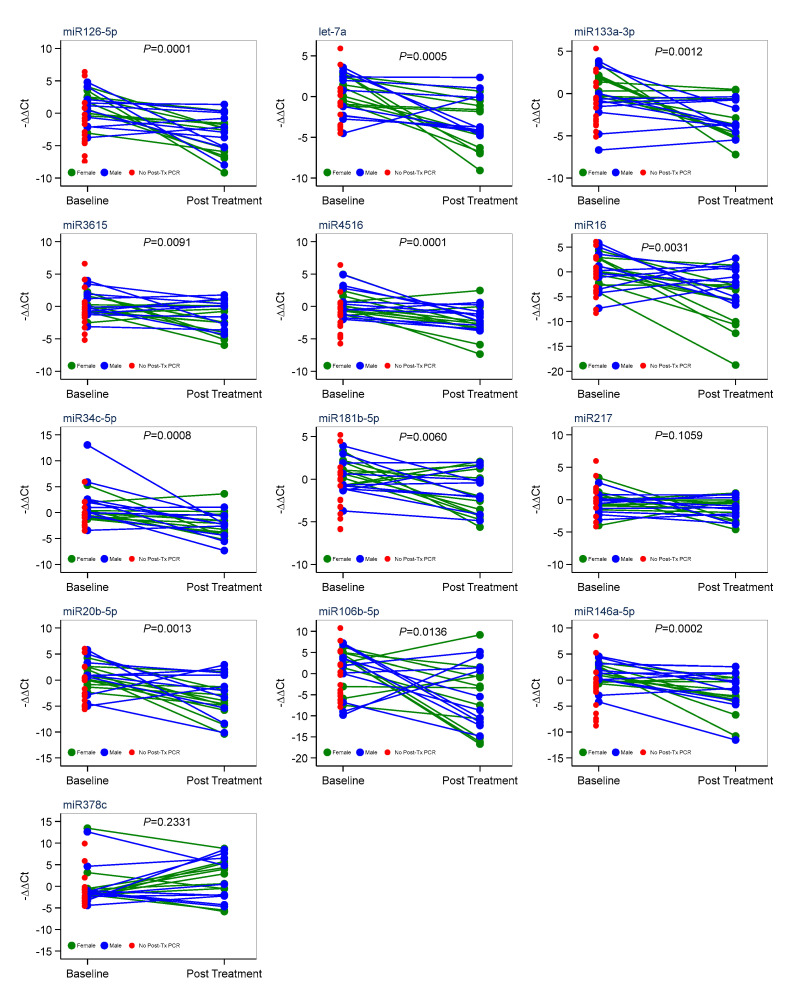
miRNA expression varying and responding to treatment in males and females with red dots representing patients who did not complete the full 90 days.

**Figure 4 cancers-16-04202-f004:**
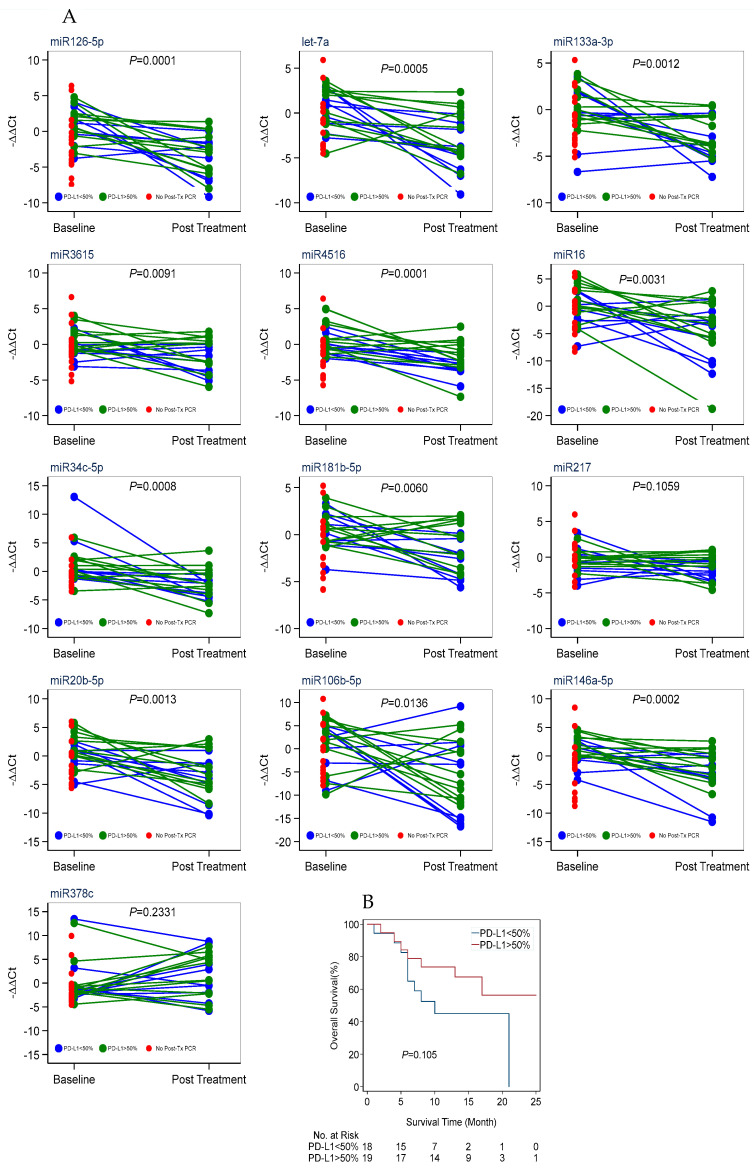
(**A**). miRNA expression varying with PD-L1 status and responding to treatment with red dots representing patients who did not complete the 90 days. (**B**). Kaplan–Meier plot showing the overall survival rate of patients from diagnosis to the last contact; comparison between PD-L1 < 50% and PD-L1 > 50%.

**Figure 5 cancers-16-04202-f005:**
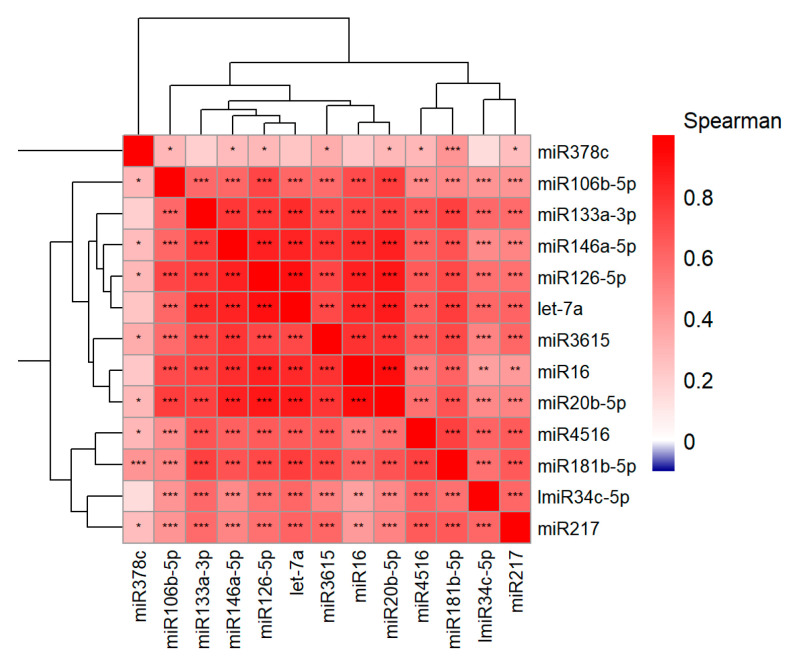
Heatmap showing correlation among miRNAs expressed in all samples. Spearman’s correlation coefficient was calculated. The coefficient ranges between −1 and 1, where −1 to 0 reflects a negative (blue) and 0 to +1 reflects a positive correlation (red), and the intensity of the shade represents the strength of the correlation. The significance of the correlation is presented in the heatmap where *** denotes *p* < 0.001, ** denotes *p* ≥ 0.001–<0.01, and * denotes *p* ≥ 0.01–<0.05.

**Table 1 cancers-16-04202-t001:** Demographic characteristics of the study cohort (*n* = 38).

Item	Descriptive Statistics
Age, y, mean (SD)	68.1 (5.62)
Gender, *n* (%)	
Female	18 (47.37)
Male	20 (52.63)
PD-L1, *n* (%)	
<50%	19 (50)
>50%	19 (50)
Smoking (packs/y), mean (SD)	38 (40.26)

**Table 2 cancers-16-04202-t002:** *p*-values of miRNA changes in overall two-way ANOVA for sex and treatment and paired *t*-tests for females and males before and after treatment.

miRNA	*p*-Value Sex	*p*-Value Treatment	*p*-Value–Males Only	*p* Value–Females Only
miR-126-5p	0.146	**<0.001**	0.058	**0.001**
Let-7a	0.087	**<0.001**	0.108	**0.002**
miR133a-3p	**0.012**	**0.001**	0.110	**0.005**
miR3615	0.446	**0.009**	0.185	**0.029**
miR4516	**0.038**	**<0.001**	**0.017**	**0.004**
miR16	0.298	**0.003**	0.405	**0.001**
miR34c-5p	0.830	**0.001**	**0.009**	**0.038**
Mi181b-5p	0.238	**0.006**	0.105	**0.031**
miR217	0.184	0.106	0.272	0.255
miR20b-5p	0.150	**0.001**	0.169	**0.001**
miR106b-5p	0.327	**0.014**	0.207	**0.033**
miR146a-5p	0.136	**<0.001**	**0.023**	**0.004**
miR378c	0.844	0.233	0.500	0.339

Statistically significant differences are in bold.

**Table 3 cancers-16-04202-t003:** Summary of survival analysis using multiple-variable Cox regression model.

Variable	Hazard Ratio (HR)	SE	Z	*p*-Value
Age	0.97	0.05	−0.57	0.568
Gender				
Female	1 (reference)			
Male	5.89	3.68	2.84	0.004
PD-L1				
<50%	1 (reference)			
>50%	0.35	0.2	−1.88	0.060
miRNA				
Let-7a	0.75	0.09	−2.51	0.012
miR217	1.45	0.24	2.21	0.027

## Data Availability

The raw data supporting the conclusions of this article will be made available by the authors on request.

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
