# Peer review of "Circulating microRNAs as a Prognostic Tool to Determine Treatment Efficacy in Lung Cancer Patients Undergoing Pembrolizumab PD-1 Blockade Immunotherapy"

_cancers, 2024, doi:10.3390/cancers16244202_

Round 1

Reviewer 1 Report

Comments and Suggestions for Authors

Reviewer Comments

Summary of the Manuscript

The manuscript titled "Circulating microRNAs as a prognostic tool to determine treatment efficacy in lung cancer patients undergoing pembrolizumab PD-1 blockade immunotherapy" evaluates the prognostic potential of circulating microRNAs (miRNAs) as biomarkers for assessing treatment efficacy in lung cancer patients treated with pembrolizumab. The authors analyzed the changes in miRNA expression levels pre- and post-treatment in a cohort of 38 patients. They further identified specific miRNAs that correlate with survival outcomes, including let-7a and miR217, potentially serving as accessible, non-invasive biomarkers. However, the manuscript requires further refinements to enhance clarity and scientific rigor.

Major strengths of the manuscript

1. Novelty in biomarker research: The study contributes to a relevant gap in cancer biomarker research by exploring the prognostic utility of circulating miRNAs in the context of PD-1 blockade immunotherapy, providing potential non-invasive methods for monitoring treatment efficacy in lung cancer.

2. Clinical relevance: The findings suggest potential applications in clinical settings, especially for patients undergoing pembrolizumab treatment, which can help predict and evaluate treatment outcomes through simple blood tests.

Major weaknesses of the manuscript

1. Language and structure: The manuscript is densely written with frequent redundant and incomplete sentence structures, impacting readability. Key sections, especially the abstract and introduction, contain redundant information, as noted in repeated points regarding PD-L1 and pembrolizumab. These should be streamlined to improve clarity.

2. Inadequate cohorts’ information: The authors do not provide sufficient demographic and clinical details on confounding factors that could influence miRNA levels, such as concurrent treatments, or other health conditions and driver mutations. Also, available research data presented by the authors lack control cohort miRNA levels evaluated in healthy individuals to highlight expression shifts specific to lung cancer.

3. Limited mechanistic insight: Although several miRNAs are identified as potential prognostic markers, their mechanistic role in PD-1 pathway modulation is not adequately addressed. Discussing the interaction of miRNAs with immune checkpoint pathways (e.g., PD-L1 expression modulation) would provide a stronger scientific basis for their prognostic value.

4. Presentation of results: Results are presented in a manner that it seems overwhelming. Figures such as Figure 1 and 5A could be restructured for readability. Data should be organized by specific themes (e.g., sex differences, PD-L1 status, survival correlation) rather than clustering multiple findings in single sections.

5. Inadequate explanation of figures and tables: Figures lack adequate descriptions, making it difficult for readers to interpret the results. Furthermore, the presentation of data in figures and tables can be improved to enhance clarity.

6. Potential bias in analysis: The study does not address potential biases in survival analysis, particularly regarding patients’ adherence to pembrolizumab treatment. A discussion on possible survival biases due to missing data points would strengthen the conclusions.

Clarity of presentation and manuscript organization

The manuscript presents a valuable dataset and explores an interesting hypothesis; however, the organization and writing could benefit from substantial improvements to enhance readability:

Abstract: The abstract does not effectively capture the significance of the research, methods, and primary findings. I recommend a concise rewrite that better highlights the study’s objective, methods, major findings, and implications for clinical practice.

Introduction: The introduction does not effectively present the background and could be condensed to avoid redundancy, especially in the discussion of PD-1 blockade. Simplify and structure the introduction to provide a clear rationale for the study, with a sequential flow. Details on PD-L1 and pembrolizumab's limitations should be discussed earlier to underscore the study’s relevance.

Results and Figures: The results section merges several points that could be distinct for better readability. Presenting figures as discrete groups (e.g., by gender or PD-L1 status, co mutation, and grade) may improve clarity. Expression of PD-L1 and associated genes should be evaluated. Pathway components and known PD-1 blockade markers should be verified for treatment veracity. Moreover, the lack of significant p-values between cohorts and proper labeling in gender-specific data analysis should be addressed to prevent misinterpretation. Separate the analysis and outline the significant p-values for clarity and avoidance of data misrepresentation.

Significant differences of miRNA in male/female pre- and post-treatment should be calculated separately, and p-values indicated. Since these miRNAs detected are not gender specific, authors should also include data showing male and female patients calculated together in each panel, separation without this would seem like authors are hiding specific important data. Further clarification of differences in response by gender can be presented in another result section. Also, these figures and tables are not properly integrated into the result.

Discussion: The discussion should expand on prior research on miRNAs and PD-1 pathways to contextualize findings. Additionally, authors should speculate on the potential of these miRNAs to predict immunotherapy response, drawing connections to previous studies where relevant.

Of note, line 249-251: This should be a premise for ERK and MEK levels (or their regulators like DUSP6, ETV, and SPRY) in these patients pre- and post-treatment. Neither was PD-L1 expression shown with the results. This opens a lot of questions about the veracity of the samples or treatment. Do authors have specific tumor mutational statuses of each patient?

Reproducibility of the manuscript

While the manuscript provides a detailed description of experimental procedures, including qPCR methods and statistical tests, the reproducibility of the study is hindered by the lack of appropriate references and access to raw data. Providing supplementary materials or access to raw data would greatly enhance reproducibility.

Constructive Comments

Expand the methods section: Include more details on miRNA selection criteria, patient demographics, mutational status, and the specific rationale behind statistical choices to improve transparency.

Cohort description and control data: Enhance demographic details and clinical variables to contextualize miRNA expression shifts. Including a control group with healthy individuals or patients with other cancers would provide contrast to lung cancer-specific miRNA changes.

Signaling pathway correlation: While the study notes miRNAs associated with survival, integrating an analysis of molecular pathways (e.g., PD-L1 modulation, RAS/MAPK) would clarify these miRNAs' roles in cancer biology and immune response. Furthermore, mRNA expression of specific modulators of immune checkpoints are needed for appropriate representation of data and proper translation of result.

Improve data presentation: Improve figure readability by separating complex data points and labeling. Ensure that separate p-values are included to reinforce statistical conclusions, especially in comparisons by gender.

Data availability statement: Providing open-access links to datasets or a statement on data availability would support reproducibility and transparency.

Accept or Reject the Manuscript + Justification

Recommendation: Major revisions

While the manuscript presents an interesting study with potential clinical implications, the issues related to cohort limitations, incomplete data, and ambiguous data presentation need to be addressed to ensure clarity and reliability of the findings. A major revision is recommended to improve transparency, enhance the clarity of data presentation, and discuss limitations more thoroughly.

Comments on the Quality of English Language

The manuscript is densely written with frequent redundant and incomplete sentence structures, impacting readability.

Key sections, especially the abstract and introduction, contain redundant information, as noted in the attached file.

Reviewer 2 Report

Comments and Suggestions for Authors

1)      I recommend the authors add numerical results to the abstract. E.g. The authors observed a significant decrease in the level of some miR in response to treatment. ( lines 32-35) How much decrease?

2)      I recommend the authors add a graphical abstract at the end of the introduction.

3)      The authors should check whether this sentence is correct or wrong! ( lines 44-45), claiming the highest level of “live”. I think it should be “mortality”

4)      What is the difference between PD-1 and apoptosis? The authors should define PD-1 before use.

5)      I suggest moving lines 77-84 to the results or conclusion since the findings/results should not be described in the introduction.

6)      Why did the authors measure miRNA expression using Kit and RT-Qpcr? They could be measured by electrochemistry or electrochemiluminescence. It is more accurate and more sensitive.

https://www.sciencedirect.com/science/article/pii/S2666831923000929

7)      It is good the authors write the mechanisms of downregulation of the investigated miRNAs after pembrolizumab treatment.

Reviewer 3 Report

Comments and Suggestions for Authors

CONRATULATIONS,NICE WORK!THE ONLY THING TO ADD IS MAYBE A PARAGRAPH OR TWO TO BE MORE DETAILED ABOUT THE CORELLATION OF miRNAs AND THE PATHWAYS OF KINASES.

Round 2

Reviewer 1 Report

Comments and Suggestions for Authors

Revisions Needed:

  1. Language and Clarity:

    • Numerous incomplete and repetitive sentences throughout the manuscript hinder readability. The authors should carefully proofread and refine the text to improve coherence and flow.
    • Example: In the introduction, some sentences merge unrelated ideas or lack proper connectors. Revise for clarity and logical sequencing.
  2. Discussion Section:

    • While comprehensive, parts of the discussion section lack focus and reiterate earlier points without adding significance. Streamlining the discussion to emphasize the novel aspects of this study would enhance impact.
  3. Data Representation:

    • The presentation of the figures, tables, and data is still unclear. For instance, the heatmaps (e.g., correlation data) could benefit from more detailed legends and annotations to aid interpretation.
    • The result section is still not fully expressive of each figure. The authors should expand the result section with references to literature on each miRNA. 
  4. Supplemental Materials:

    • The demographics and additional methodological details provided in the supplemental materials should be briefly summarized in the main text for better context.
  5. References and Citations:

    • Ensure uniformity in formatting references and avoid mixing outdated studies with more recent, relevant literature.
Comments on the Quality of English Language
  • Numerous incomplete and repetitive sentences throughout the manuscript hinder readability. The authors should carefully proofread and refine the text to improve coherence and flow.
  • Example: In the introduction, some sentences merge unrelated ideas or lack proper connectors. Revise for clarity and logical sequencing.

Reviewer 2 Report

Comments and Suggestions for Authors

The MS has been improved and I recommend to be published. 

Author Response

Dear reviewer,

Thank you for recommending our revised manuscript for publication. We are grateful for your feedback and comments, and it greatly helped in improving the manuscript.

Thank you.